# OpenReview forum: "Diffusion of Thought: Chain-of-Thought Reasoning in Diffusion Language Models"
_NeurIPS.cc/2024/Conference — NeurIPS 2024 poster_

### Official Review · Reviewer_q6Ny · 2024-07-06

**Soundness:** 3
**Presentation:** 3
**Contribution:** 3
**Rating:** 7
**Confidence:** 4

**Summary:**

The paper introduces Diffusion of Thought （DoT）, integrating diffusion models with Chain-of-Thought. The paper proposes two training-time sampling strategies to enhance self-correction during inference. Experimental results demonstrate the effectiveness of DoT in simple and complex reasoning tasks.

**Strengths:**

- The paper presents an initial exploration into the reasoning ability of current diffusion language models.
- In simple reasoning tasks, DoT achieves up to 27× speed-up without performance drop compared to CoT and implicit CoT.
- DoT showcases promising self-correction abilities in complex reasoning problems.

**Weaknesses:**

- The coupled sampling strategy, designed to rectify errors in previous thoughts, appears to assume that the noise added to the rationale $r_{i-k}, \cdots, r_{i-1}$ is the same as the potential errors in previous rationales during inference. This assumption is not intuitively obvious and lacks a clear explanation.
- Despite building upon the DiffuSeq framework, the paper does not include comparisons with the DiffuSeq model, which has the most similar model backbone.
- It would be better if the paper could provide a qualitative comparison of the reasoning paths between DoT and DoT$^{MP}$

**Questions:**

- L128-129 is confusing, and it is unclear how Table 2 supports the statement that “the gradient-based token guidance fails to do accurate conditioning as the model cannot exactly recover each conditioning token”.
- L153-154, it says that the model “mimics the inference stage with probability $\epsilon_i$”, and “$\epsilon_i$ linearly decays from 1 to $\epsilon_{min}$”. However, this suggests that the model utilizes $\hat{z}_o = z_θ(z_u;u)$ at the start of training. Given this setup, when training from scratch, would there be concerns that $z_θ()$ would fail to predict a meaningful $\hat{z}_0$ at the beginning of training.?
- How does the model decide on the number of rationales in the multi-pass DoT?

**Limitations:**

The paper has adequately stated the limitations.

---

> ### Author Rebuttal · Authors · 2024-08-06
>
> We sincerely thank Reviewer q6Ny for the review and are grateful for the time you spent with our submission. We wish to address your confusion and concerns by providing detailed responses to each of your comments.
>
> **Weakness 1: Confusion about the coupled sampling strategy**
>
> Thanks for pointing out this potential confusion. The main purpose of the coupled sampling mechanism is to enable the DoT$^{MP}$ model with the ability to correct potential errors in the previous thoughts. Without coupled sampling, the discrepancy arises between the use of correct previous thoughts during training and the possibly erroneous generated thoughts during testing, leading to error accumulation akin to that in autoregressive models. To alleviate this issue, we introduce the coupled sampling strategy for the model to learn to rectify past errors. This strategy involves injecting noise into previous thoughts during the training phase, enabling the model to possess the capability to see and correct errors in previous thoughts. We will add more details to clarify this confusion in the final version.
>
> **Weakness 2: Comparisons with the DiffuSeq model**
>
> Thank you for bringing up this point. We have this comparison in Table 2 and here is the summary of the comparison between DiffuSeq and DoT. We will clarify this confusion in the final version.
>
> |                  | Accuracy |
> |------------------|----------|
> | Plaid + DiffuSeq | 31.2     |
> | Plaid + DoT      | 32.6     |
> | Plaid + DoT$^{MP}$ | 37.7     |
>
> **Weakness 3: Qualitative comparison of the reasoning paths between DoT and DoT$^{MP}$**
>
> Thank you for your suggestion. We observe that DoT$^{MP}$ outperforms DoT in correctness regarding the reasoning paths, while DoT slightly excels in diversity as depicted in Figure 4(b). Below we show some examples where DoT$^{MP}$ can predict the correct reasoning path while DoT fails. More content related to the reasoning path analysis will be incorporated in the paper accordingly.
>
> >*Query*: The Kennel house keeps 3 German Shepherds and 2 Bulldogs. If a German Shepherd consumes 5 kilograms of dog food and a bulldog consumes 3 kilograms of dog food per day. How many kilograms of dog food will they need in a week?
> >
> >*DoT*: <<3*5=15>> <<7*3=21>> <<15+21=36>> #### 36
> >
> >*DoT$^{MP}$*: <<3*5=15>> <<2*3=6>> <<15+6=21>> <<21*7=147>> #### 147
>
> > *Query*: Skyler has 100 hats on his hand with the colors red, blue, and white. Half of the hats are red, 3/5 of the remaining hats are blue, and the rest are white. How many white hats does Skyler have?
> >
> >*DoT*: <<1/2*100=50>> <<3/5*50=30>> <<100-30=70>> #### 70
> >
> >*DoT$^{MP}$*: <<100/2=50>> <<100-50=50>> <<50*3/5=30>> <<50-30=20>> #### 20
>
> **Question 1: Confusion about L128-129**
>
> Thanks for pointing out this potential confusion. The first line of Table 2 is the tuned Plaid using the gradient-based token guidance to generate response and it achieves poor performance. In Plaid, the use of gradient-based guidance to inject conditions involves adjusting random source embeddings through gradients to match condition tokens. However, we observe discrepancies between recovered source tokens and condition tokens, which can adversely affect tasks requiring precise conditioning. Below we show an example on grade school math as a demonstration, where **bold** words in the query part are incorrectly recovered. We can see there are three recovered query tokens that exhibit minor differences due to soft gradient guidance, causing interference with the model's comprehension of the problem. That’s why we resort to hard control with gradient-free conditioning. We will add more details to clarify this confusion in the final version.
>
> > *Groundtruth*: Two trains leave San Rafael at the same time. They begin traveling westward, both traveling for 80 miles. The next day, they travel northwards, covering 150 miles. What's the distance covered by each train in the two days? <<2*80=160>> <<150*2=300>> <<300+160=460>> <<460/2=230>> #### 230
> >
> > *Prediction*: **Three** trains leave San Juan at the same time. They **start** traveling westward, both traveling for 80 miles. The next day, they travel **southward**, covering 150 miles. What's the distance covered by each train in the two days? <<3*80=180>> <<180+80+150=340>> <<340/ 30=12.5>> #### 12.5
>
>
> **Question 2: About probability $\epsilon$**
>
> Thank you for bringing up this excellent question. On one hand, we actually desire the presence of noise in $z_0$, as it forces the model to pose the self-correction ability. On the other hand, due to our small $\epsilon_\text{min}$, i.e., 0.95, we primarily rely on gold data, thereby preventing excessive noise in $z_0$ that could make training too challenging. We previously attempted a warmup process by integrating scheduled sampling after a certain number of steps, but we did not observe significant performance improvements. Hence, for the sake of simplicity, we opted for the current approach.
>
> **Question 3: How does the model decide on the number of rationales**
>
> Thank you for bringing up this point. During training, we append a special token <EOS> to the last thought, so when the model generates a thought followed by <EOS>, it stops generating further. We will add this detail to our manuscript.

---

> > ### Author Response · Authors · 2024-08-10
> >
> > Dear Reviewer q6Ny,
> >
> > Thank you for your valuable time to review our work and for your constructive feedback. We posted our response to your comments two days ago, and we wonder if you could kindly share some of your thoughts so we can keep the discussion rolling to address your concerns if there are any. If you have any further questions, we are happy to discuss them!
> >
> > Best regards,
> >
> > Authors

---

> > > ### Comment · Reviewer_q6Ny · 2024-08-12
> > >
> > > Thank you for your response and my apologies for the delayed reply. I still have a few questions that I hope the authors can clarify.
> > >
> > > W1: I understand that the purpose of coupled sampling is to correct errors in previous inferences. However, I feel like the noise introduced during training is not essentially the same as the potential errors during inference. Therefore, it’s not clear to me how this approach equips the model to identify and rectify the errors.  Also, I wonder if the model needs to inject noise into previous thoughts during inference as well? Moreover, how is this noise added during training—is it uniformly applied to all previous $i-k$ rationales, and how much noise needs to be injected?
> > >
> > > W2: Table 2 provides a comparison of Plaid + DiffuSeq and Plaid + DoT using only the GSM8K dataset. Given that the main novelty of the paper is the DoT idea and the proposed two sampling strategies. could you extend the comparison to include more datasets? Specifically, a comparison involving Plaid CoT vs. Plaid DoT and SEDD CoT vs. SEDD DoT, all employing the DiffuSeq training method, would be valuable to demonstrate the effectiveness of the proposed methods. My concern is about how much performance gain is attributed to the DiffuSeq training method (i.e., not using the gradient-based guidance.)

---

> > > > ### Author Response · Authors · 2024-08-14
> > > > **Reply to Further Question from Reviewer q6Ny**
> > > >
> > > > We really appreciate Reviewer q6Ny for your effort in reviewing our paper and thanks for your insightful questions.
> > > >
> > > > **Q1: Therefore, it’s not clear to me how this approach equips the model to identify and rectify the errors.**
> > > >
> > > > Yes. The discrepancy exists between the noise present during training and the potential errors during inference. The reason why our coupled sampling would work is to consider that DoT$^{MP}$ might generate the incorrect previous rationale. Without coupled sampling, DoT$^{MP}$ may exhibit behavior similar to that of the causal language model, leading to error accumulation. In contrast, by incorporating coupled sampling, DoT$^{MP}$ is able to effectively learn how to recover the correct rationale $r_{i}$ despite errors (noises) in previous rationales. About the discrepancy, we have another explanation in the following part.
> > > >
> > > > **Q2: if the model needs to inject noise into previous thoughts during inference as well?**
> > > >
> > > > We have tried to inject the noise into the previous thoughts. The implementation is quite easy with the controlled source mask. During the sampling process, noise is only injected for previous thoughts with a timestep smaller than a specific threshold. This ensures that the previous thoughts remain meaningful rather than being influenced by random noise when the timestep is larger. The experiment shows that this setting does not bring significant improvements so we didn’t use this in the end. We will add these experiments to the Appendix in the next version.
> > > >
> > > > **Q3: how is this noise added during training—is it uniformly applied to all previous 𝑖−𝑘 rationales, and how much noise needs to be injected?**
> > > >
> > > > We employ the same noising strategy as learning the next thought for previous i-k thoughts, i.e., Gaussian noise for continuous diffusion or categorical noise for discrete diffusion. As mentioned in the paper, we will only select a very small portion of the training data to add this noise, to make sure the training stability.
> > > >
> > > > **Q4: the effect of DiffuSeq-style training and using more datasets**
> > > >
> > > > Thanks for the suggestion. We only chose the GSM dataset because the accuracy results are distinguishable. For bool logic and digit multiplication tasks, the accuracy performance is good enough and we focus on the efficiency of DoT. We will add more ablation datasets to further support our findings. We notice the performance gain of employing the DiffuSeq training method is significant compared to the default continuing pretraining paradigm (i.e., without freezing the condition tokens) for both Plaid and SEDD (see Tables below). Our DoT builds on DiffuSeq and further enhances the performance by incorporating strategic sampling methods and the multi-pass reasoning paradigm. We will update detailed experiments and additional datasets in the Appendix in the next version.
> > > >
> > > > |                              | Accuracy | Accuracy Gain |
> > > > |------------------------------|----------|---------------|
> > > > | Plaid + continue pretraining | 0.5      | -             |
> > > > | Plaid + DiffuSeq-style training             | 31.2     | +30.7         |
> > > > | Plaid + DoT                  | 32.6     | +32.1         |
> > > > | Plaid + DoT$^{MP}$             | 37.7     | +37.2         |
> > > >
> > > > |                             | Accuracy | Accuracy Gain |
> > > > |-----------------------------|----------|---------------|
> > > > | SEDD + continue pretraining | 5.9      | -             |
> > > > | SEDD + DiffuSeq-style training | 32.1     | +26.2         |
> > > > | SEDD + DoT                  | 34.8     | +28.9         |
> > > > | SEDD + DoT$^{MP}$             | 35.0     | +29.1         |

---

### Official Review · Reviewer_Y1KX · 2024-07-11

**Soundness:** 2
**Presentation:** 3
**Contribution:** 2
**Rating:** 5
**Confidence:** 4

**Summary:**

This paper introduces "Diffusion of Thought" (DoT) to diffusion language models to improve upon their reasoning capabilities.

The method adapts the implicit chain of thought methodology (iCoT) for autoregressive models, which relies on per-task fine-tuning to distill reasoning into transformer layers, while DoT encodes it into diffusion steps.

The methodology includes a comparison between a single-pass and a multi-pass approach. The single-pass averages all rationales across all timesteps, while the multi-pass introduces a causal inductive bias between rationales by averaging each reasoning step at a time across all timesteps.

Similar to the iCoT paper, evaluations are conducted on tasks lmultiplication, boolean logic, and the grade school math (GSM8K) tasks.

The approach leverages self-correction, self-consistency, and the number of reasoning steps (T) to further improve accuracy, trading off some efficiency.

**Strengths:**

- The fundamental idea of encoding reasoning rationales into diffusion steps seems an intuitive path to explore.
-  Due to the flexible timestep parameter (T), DoT offers greater flexibility compared to Implicit Chain of Thought (iCoT), which is limited by the number of transformer layers.

**Weaknesses:**

- **Direct Comparison Baseline** The paper lacks a direct comparison with answer-only and  traditional CoT technique applied to diffusion language models, which would provide a clearer benchmark for evaluating the effectiveness of DoT. The paper only provides a comparison with auto-regressive answer-only, CoT and iCoT results. This does not convincingly demonstrate that the additional complexity introduced is justified by performance improvements.
  - The paper does not adequately separate between the specific contributions of the DoT methodology from the inherent advantages of using diffusion language models.

- **Missing iCoT context** Section 3 does not clearly explain how DoT builds on the implicit Chain-of-Thought (iCoT) approach, especially regarding the training operations. Instead, it focuses mainly on additional complexities and mechanisms introduced to improve overall results. A detailed connection between iCoT and DoT is needed to better understand the modifications and their impact. The terms 'single-pass' and 'multi-pass' could be misleading as they typically imply batch processing. Here, they refer to how probabilities of different reasoning paths are handled, in parallel or sequentially.

- **Task-Specific Fine-Tuning Requirement**  DoT performs well on simple tasks like multiplication but requires fine-tuning with a larger number of reasoning steps t for grade school math. This contrasts with CoT methods in autoregressive models, which can adapt more flexibly using examples directly in the input.

- **Throughput Comparison** The absence of a direct throughput comparison for fixed **T** across evaluation settings limits understanding of T's impact on performance and efficiency. The results in Table 1 summarized results for dynamic sampling timesteps T.

**Questions:**

Could the authors clarify why they chose to compare DoT with answer-only, CoT, and iCoT for autoregressive models, but did not include similar comparisons with answer-only and CoT for diffusion language models as well?

**Limitations:**

yes, the authors acknowledge the reliance on specialized training per reasoning task and the limited generalization capabilities.
The need for more reasoning steps as tasks become more complex could be further elaborated upon.

---

> ### Author Rebuttal · Authors · 2024-08-06
>
> We sincerely thank Reviewer Y1KX for your review and are grateful for the time you spent on our submission. Below we would like to give detailed responses to each of your comments.
>
> **Weakness 1: Direct Comparison Baseline**
>
> Thank you for your suggestion. We conduct the answer-only setting to further validate the effectiveness of DoT. The result table reveals that fine-tuning diffusion models solely with answer data leads to inferior performance compared to DoT, mirroring the degradation of AR models in the absence of CoT.
>
> |                   | Accuracy |
> |-------------------|----------|
> | GPT-2 Answer-only | 17.0     |
> | GPT-2 CoT         | 43.9     |
> | Plaid Answer-only | 12.4     |
> | Plaid DoT         | 37.7     |
> | SEDD Answer-only  | 29.1     |
> | SEDD DoT          | 45.7     |
>
> > "The paper does not adequately separate between the specific contributions of the DoT methodology from the inherent advantages of using diffusion language models."
>
> Thank you for bringing up this point. Firstly, one of our contributions is exactly employing diffusion language models for multi-step text reasoning. To the best of our knowledge, we are the first to bring diffusion into the realm of complex text reasoning such as mathematical reasoning. Additionally, we show fine-tuning a pretrained diffusion model is a non-trivial work. As demonstrated in the ablation study (Table 2), directly following the pretraining approach leads to subpar results. Therefore, we resorted to an infilling approach and further proposed a series of sampling strategies and multipass variants to enhance the model's performance. The comparison between applying CoT and DoT to diffusion is presented below.
>
> |                   | Accuracy |
> |-------------------|----------|
> | Plaid CoT  (gradient-based guidance) | 0.5  |
> | Plaid CoT                            | 31.2 |
> | Plaid DoT$^{MP}$                       | 37.7 |
>
> **Weakness 2: Missing iCoT context**
>
> Thank you for your suggestion about more description about iCoT. In Section 3.1, we list 3 parallel approaches of modeling CoT: AR, iCoT, DoT, theoretically. Below, we discuss some similarities and differences between iCoT and DoT. DoT shares similarities with iCoT in the following 3 high-level aspects: i) Both DoT and iCoT try to tackle the time cost of auto-regressively generating the chain-of-thought rationales; ii) Both DoT and iCoT process “thoughts” “vertically’’ in hidden dimension. But DoT presents the hidden information across different diffusion timesteps (in temporal dimension), while iCoT presents the hidden information across the model’s different layers (in spatial dimension); iii) Both DoT and iCoT evaluate the model’s CoT ability, so we refer to some experiment settings of iCoT, including datasets and baselines.
>
> However, in terms of methodology, iCoT and DoT are completely different. iCoT still relies on next-token prediction for autoregressive generation, while DoT utilizes diffusion, which offers additional advantages such as a flexible performance-efficiency trade-off and self-correction capability beyond efficiency. We will add more clarification about DoT and iCoT in the paper.
>
> Thanks for pointing out the potential confusion about the names of ‘single pass’ and ‘multi-pass’, we will clarify them in the paper.
>
>
> **Weakness 3: Task-Specific Fine-Tuning Requirement**
>
> Thank you for your constructive comments. The main reason is that the current pre-trained diffusion language models are relatively small, resulting in the underutilization of their in-context learning capabilities, as we discussed in the limitations. Exploring in-context learning for text diffusion is another interesting topic.
>
> **Weakness 4: Throughput Comparison**
>
> Thank you for your suggestion. In Appendix L713-L714, we provide a detailed description on T for in Table 1. Specifically, we utilize T = 1 for digit multiplication, T = 2 for the boolean logic dataset, and T = 64 for grade school math. It is worth noting that adjusting the parameter T itself is an advantage of DoT, as it allows us to allocate computational resources more efficiently to challenging tasks, while using a smaller T for simpler tasks. We show how T affects performance on grade school math in Figure 3, and below we also show how T affects throughput for Plaid DoT$^{MP}$. The relationship between throughput and T appears to be nearly linear.
>
> | T   | Accuracy | Throughput |
> |-----|----------|------------|
> | 1   | 18.18    | 6.6        |
> | 2   | 35.9     | 3.4        |
> | 4   | 36.7     | 1.7        |
> | 8   | 36.4     | 0.9        |
> | 16  | 36.1     | 0.4        |
> | 32  | 37.4     | 0.2        |
> | 64  | 37.7     | 0.1        |
> | 128 | 37.7     | 0.05       |
>
> **Q1: comparisons with answer-only and CoT for diffusion language models**
>
> Please see the response for weakness 1.
>
> **Limitation: The need for more reasoning steps as tasks become more complex could be further elaborated upon.**
>
> Thank you for your suggestion. The main point of the CoT paper is to improve the reasoning ability by involving more intermediate steps. In DoT, we also observed that more diffusion timesteps (computing FLOPs) yield better results. [1] also presents a similar idea, and this topic is worthy of further investigation.
>
> [1] Pondernet: Learning to ponder. ICML 2021.

---

> > ### Author Response · Authors · 2024-08-10
> >
> > Dear Reviewer Y1KX,
> >
> > Thank you for your valuable time to review our work and for your constructive feedback. We posted our response to your comments two days ago, and we wonder if you could kindly share some of your thoughts so we can keep the discussion rolling to address your concerns if there are any. If you have any further questions, we are happy to discuss them!
> >
> > Best regards,
> >
> > Authors

---

> > ### Comment · Reviewer_Y1KX · 2024-08-12
> >
> > Thank you for the detailed rebuttal and for providing the necessary comparisons. These results should have been included in the initial submission to support the narrative of the paper. Given these updates, I will increase the current score to 5.

---

> > > ### Author Response · Authors · 2024-08-14
> > >
> > > Dear Reviewer Y1KX,
> > >
> > > Thank you for your approval of our work. We sincerely appreciate your suggestions to enhance the rigor of our paper. We would be happy to do any follow-up discussion or address any additional comments.
> > >
> > > Best regards,
> > >
> > > Authors

---

### Official Review · Reviewer_5GKr · 2024-07-11

**Soundness:** 3
**Presentation:** 2
**Contribution:** 3
**Rating:** 5
**Confidence:** 3

**Summary:**

The authors propose a chain-of-thought technique for diffusion language models. They achieve this by diffusing a set of hidden representations (thoughts) through time. Different sampling techniques are introduced to enhance error recovery including looking forward and conditioning on multiple previous thought steps in predicting the current thought. They achieve competitive results in terms of throughput compared to chain-of-thought paradigms applied to small language models.

**Strengths:**

- The authors extend the chain-of-thought paradigm to language diffusion models, which is novel and significant.
- Their results seem to suggest that this is a promising direction.

**Weaknesses:**

- The presentation can be enhanced:
  - The transparent figure colors are very hard to read.
  - The figures do not render correctly on different PDF viewers.
- Comparison to larger open language models (e.g., Lama) would improve this contribution's placement in the literature.

**Questions:**

I believe that in line 155, the first word should be "future" instead of "former," and the last word should be "backward" instead of "forward." Is this a typo or a misunderstanding on my part?
- Additionally, I think a comparison to the paradigm in [1] could be informative.


[1] Harvey W, Wood F. Visual chain-of-thought diffusion models. arXiv preprint arXiv:2303.16187. 2023 Mar 28.

**Limitations:**

The authors discussed the limitations.

---

> ### Author Rebuttal · Authors · 2024-08-06
>
> We sincerely thank Reviewer 5GKr for your review and are grateful for the time you spent on our submission. We are also glad you think our paper is novel and significant. Below we would like to give detailed responses to each of your comments.
>
> **Weakness 1: The presentation regarding color and figure rendering can be enhanced**
>
> Thank you for bringing to our attention the potential confusion regarding color and figure rendering. We will address and clarify this issue in the final version of the paper.
>
> **Weakness 2: Comparison to larger open language models (e.g., Llama)**
>
> Thank you for your suggestion. We add the results for (LoRA) fine-tuning LLMs on the same dataset, listed in the following table. Please note that the current diffusion pretrained model is much smaller than Llama 7B, so this comparison is not fair and we just list them for reference. We have validated that our DoT is better than the same scale autoregressive model GPT-2 (Table 1), which shares the similar architecture with Llama. We believe that further exploration of diffusion language models will lead to larger models that can compete with current LLMs, allowing DoT to achieve results more comparable to Llama.
>
> |                 | Params | Accuracy |
> |-----------------|--------|----------|
> | GPT-2 CoT       | 355M   | 43.9     |
> | Mistral CoT     | 7B     | 68.8     |
> | Llama CoT       | 7B     | 59.0     |
> | SEDD DoT (Ours) | 424M   | 45.7     |
>
> **Q1: About the confusion in line 155**
>
> Thank you for bringing up this question. For the first “former”: in the inference stage of diffusion, the timestep t starts from T and progressively decreases to 1, so we refer the larger time $u$ to “former” steps. Regarding two “forward” words in this line, this term carries different meanings: the first “forward” refers to the forward process in diffusion, which involves adding noise to data, while the last “forward” denotes the forward pass of the model, in contrast to the backward gradient-backpropagation pass. We will avoid using the same word with different meanings to prevent misunderstandings.
>
> **Q2: A comparison to the paradigm in [1] could be informative.**
>
> Thank you for sharing this paper ‘Visual CoT of diffusion models’. This paper borrows the idea of CoT in LLMs which involves intermediate steps to improve the performance. Their model acts in two steps: first generate the CLIP embedding and then generate the final image. This paper and our DoT both mentioned the CoT in diffusion models, but there is a big difference: the mentioned paper only borrows the idea of CoT but cannot perform the CoT reasoning, while DoT focuses on the reasoning ability of text models, as the alternative to autoregressive CoT in LLMs. We will add this comparison in related work.

---

> > ### Author Response · Authors · 2024-08-10
> >
> > Dear Reviewer 5GKr,
> >
> > Thank you for your valuable time to review our work and for your constructive feedback. We posted our response to your comments two days ago, and we wonder if you could kindly share some of your thoughts so we can keep the discussion rolling to address your concerns if there are any. If you have any further questions, we are happy to discuss them!
> >
> > Best regards,
> >
> > Authors

---

> > > ### Author Response · Authors · 2024-08-14
> > >
> > > Dear Reviewer 5GKr,
> > >
> > > Thank you for your valuable time to review our work and constructive feedback. As the discussion period draws to a close, we would appreciate it if you could kindly take a look at our response to your comments. If you have any further questions, we are happy to discuss them!
> > >
> > > Thanks very much!
> > >
> > > Best regards,
> > >
> > > Authors

---

> > > > ### Author Response · Authors · 2024-08-14
> > > >
> > > > Dear Reviewer 5GKr,
> > > >
> > > > As the discussion period draws to a close, we would like to emphasize the main point of our paper for your reference. Most reasoning paradigms currently rely heavily on CoT with autoregressive language models. The question remains as to whether diffusion language models are capable of reasoning like AR with CoT or surpass the performance of AR with CoT in efficiency or performance. In this work, we introduce DoT as an initial exploration in this direction and demonstrate the promising potential of diffusion models for reasoning tasks.
> > > >
> > > > Best regards,
> > > >
> > > > Authors

---

### Official Review · Reviewer_E7FJ · 2024-07-19

**Soundness:** 3
**Presentation:** 2
**Contribution:** 3
**Rating:** 5
**Confidence:** 2

**Summary:**

The work introduces Diffusion-of-Thought (DoT), a method that combines diffusion language models with the Chain-of-Thought technique to enhance their reasoning ability. DoT uses the flexibility of diffusion processes to allow reasoning steps to diffuse over time, improving performance in several mathematical tasks, and demonstrating its self-correction abilities. The experimental results show DoT's effectiveness in many tasks.

**Strengths:**

- The work deals with an important problem in ML, verifying reasoning ability on a recently arisen diffusion language model.
- The proposed method is technically sound.
- The experiments show DoT's empirical effectiveness on many math benchmarks.

**Weaknesses:**

- Some of the recent work is not discussed [1]
- No standard deviation or confidence interval in the results.

---

[1] Can mamba learn how to learn? a comparative study on in-context learning tasks, ICML 2024

**Questions:**

- In Figure 2, can you represent the rationale example in the DoT chart in a similar way to the Problem-solving tasks chart on the left (like 2+1=3 in the grey box)? What exactly is the rationale for the DoT chart?
- Is the performance improvement attributed to enhanced reasoning? Could it simply be due to fine-tuning? It would be beneficial to compare the results with a method that has been fine-tuned without using DoT.

**Limitations:**

- Limited ablation study
- General performance improvement beyond mathematical tasks are not discussed

---

> ### Author Rebuttal · Authors · 2024-08-06
>
> We sincerely thank Reviewer E7FJ for your review and are grateful for the time you spent on our submission. We're pleased you find our method effective. Below, we provide a point-by-point rebuttal to clarify your concerns.
>
> **Weakness 1: Discussion of recent work**
>
> Thank you for sharing this paper *Can mamba learn how to learn?*. We noticed that there are several differences between this paper and DoT. First, mamba is a new model architecture as an alternative to traditional Transformers with full attention. For our diffusion models, we currently use traditional Transformers architecture, and it is orthogonal to the design of mamba. It would be interesting to see DoT’s performance with mamba as the base model. Second, the mamba paper mainly discusses the ability of in-context-learning, while our experiment setting focuses on the chain-of-thought reasoning. In all experiments except for few-shot ChatGPT baselines, we didn’t use in-context demonstrations. Exploring ICL ability of diffusion models is another interesting topic.
>
> **Weakness 2: Standard deviation**
>
> Thank you for bringing up this point. All experimental results were obtained by averaging the results of 3 separate trials, with a confidence interval of p < 0.01. Experimental results also reveal significant disparities in accuracy among different models.
>
> **Question 1: Rationale example in Figure 2**
>
> Thank you for bringing up this question. In the problem-solving tasks chart, we have two rationales and one final answer, 3 CoT steps in total `<<2/2=1>><<2+1=3>>####3`. For AR models, it will generate each token **one by one**. For single-pass DoT, it will generate the whole CoT steps in parallel: `<<2/2=1>><<2+1=3>>####3` (Table 3).  For multi-pass DoT, it will generate `<<2/2=1>>` first in parallel, then `<<2+1=3>>`, and then `####3`.
>
> **Question 2: Comparison with no-DoT finetune**
>
> Thank you for your suggestion. We conduct the answer-only setting to further validate the effectiveness of DoT. The result table reveals that fine-tuning diffusion models solely with answer data leads to inferior performance compared to DoT, mirroring the degradation of AR models in the absence of CoT.
>
> |                  | Accuracy |
> |------------------|----------|
> |GPT-2 Answer-only|17.0|
> |GPT-2 CoT | 43.9 |
> |Plaid Answer-only |12.4|
> |Plaid DoT | 37.7|
> |SEDD Answer-only|29.1|
> |SEDD DoT|45.7|
>
> **Limitation: Ablation study and general performance**
>
> Thank you for your suggestion. We have listed the comparison with the no-DoT fine-tune and will add it to the paper. The current ablation experiments further validate DoT’s effectiveness.
>
> In this work, we mainly focus on the reasoning ability of models including both logical reasoning and mathematical reasoning. For more general testing like performing a variety of complex tasks such as for ChatGPT, further advancements are still required to enhance the scalability of diffusion language models for general ability, as described in our limitation section.

---

> > ### Author Response · Authors · 2024-08-10
> >
> > Dear Reviewer E7FJ,
> >
> > Thank you for your valuable time to review our work and for your constructive feedback. We posted our response to your comments two days ago, and we wonder if you could kindly share some of your thoughts so we can keep the discussion rolling to address your concerns if there are any. If you have any further questions, we are happy to discuss them!
> >
> > Best regards,
> >
> > Authors

---

> > > ### Author Response · Authors · 2024-08-14
> > >
> > > Dear Reviewer E7FJ,
> > >
> > > Thank you for your valuable time to review our work and constructive feedback. As the discussion period draws to a close, we would appreciate it if you could kindly take a look at our response to your comments. If you have any further questions, we are happy to discuss them!
> > >
> > > Thanks very much!
> > >
> > > Best regards,
> > >
> > > Authors

---

> > > > ### Comment · Reviewer_E7FJ · 2024-08-14
> > > >
> > > > Thank you very much to the authors for their detailed response. I've read all the comments and other reviewer's concerns. While the rebuttal clarified most of my questions, I still have concerns about the presentation and think it needs improvements. I will keep my score.

---

> > > > > ### Author Response · Authors · 2024-08-14
> > > > >
> > > > > Dear Reviewer E7FJ,
> > > > >
> > > > > We appreciate your valuable feedback and suggestions. We will add all necessary details to our manuscript. We would like to emphasize the main point of our paper for your reference. Most reasoning paradigms currently rely heavily on CoT with autoregressive language models. The question remains as to whether diffusion language models are capable of reasoning like AR with CoT or surpass the performance of AR with CoT in efficiency or performance. In this work, we introduce DoT as an initial exploration in this direction and demonstrate the promising potential of diffusion models for reasoning tasks.
> > > > >
> > > > > Best regards,
> > > > >
> > > > > Authors

---

### Official Review · Reviewer_abfr · 2024-07-22

**Soundness:** 3
**Presentation:** 3
**Contribution:** 2
**Rating:** 6
**Confidence:** 3

**Summary:**

* The authors propose DoT, a chain of thought method for diffusion language models.
* DoT is applicable to both continuous embedding-based diffusion models and continuous-time Markov chain discrete diffusion models.
* DoT shows performance increase on digit multiplication, boolean logic, and GSM8K tasks, as well as tradeoffs in reasonability and efficiency.
* Overall, this is a relevant work in the growing field of diffusion language modeling that applies CoT reasoning from the AR literature.

**Strengths:**

* DoT is applied to both discrete and continuous diffusion language models. Given that there are various formulations of diffusion language models (embedding diffusion, simplex diffusion, masking state / absorption, continuous-time Markov chain), this is a plus.
* The authors also demonstrate DoT both by pretraining small models (standard 12-layer transformer with 6 encoder and decoder layers, respectively) from scratch, as well as leveraging pretrained diffusion models (Plaid, SEDD).
* DoT shows strong performance across multiplication, boolean, and GSM8K datasets, outperforming GPT-2 baselines.
* Empirically, the authors demonstrate that it is possible for diffusion models to have flexibly thought processes, where the model builds off of intermediate thoughts to arrive at the correct answer (similar to AR CoT) or jump to an answer, then correct its intermediate steps.

**Weaknesses:**

* The dataset explored in this work seems rather simple. Although the work understandably builds on top of previous work that employs the same dataset, the fact that baseline models achieve 100% or close to 100% makes it difficult to lucidly compare the baseline with the proposed approach. This applies to both multiplication setups as well as boolean logic, where GPT-2 models already reach 100% even without CoT.
* The authors use throughput as the basis for why DoT is superior to AR CoT when both methods achieve 100%. This is a slightly weaker argument because throughput for diffusion models critically depends on a number of hand-crafted parameters, such as the number of backward steps and the model context length. These parameters are orthogonal to DoT. It is possible that the particular setup of this work was favorable to diffusion, but not necessarily so in the general case. Moreover, AR models can leverage key-value caching to speed up generation, whereas diffusion models cannot. I am not sure if I am entirely convinced that DoT would generally be faster than CoT in the wild.
* Although CoT with diffusion models is a new area, the methodology itself is not entirely novel, as it appears to be an adaptation of DiffuSeq-style masking applied to CoT training data (i.e., give the model a question as its prefix context, and diffuse over the answer + CoT intermediate steps).

**Questions:**

* Could you quickly clarify what you mean in the first sentence of Section 3.2? I was not able to find a direct connection between the motivating claim and Table 2.
* Did you notice a big difference in the quality or diversity of the diffusion model output when softmax temperature smoothing was not applied?
* Do you train the model to predict padding tokens so that it can output sequences of variable lengths? If so, does the diffusion model always generate 128 tokens?
* Scheduled sampling essentially uses the prediction from the previous timestep instead of the noised ground truth as the condition to the diffusion model (similar to how teacher forcing is stochastically applied when training autoregressive models). This seems like a variation of self-conditioning [1, 2, 3], which incorporates model predictions from the previous timestep to generate predictions at the current timestep. It would be instructive to delineate any similarities and differences between self-conditioning and the proposed scheduled sampling.
* In the GPT-2 throughput benchmarks, did you enable K-V caching, flash attention, and other standard techniques for accelerating the forward pass?
* Table 3 demonstrates the GSM8K CoT format used in this work. Did you preprocess the dataset to strip all natural language and extract only `<< blah >>` expressions? If not, did you find that the model was able to generate coherent natural language expressions that "made sense" along with the equations and the final answer?

---
[1] Self-conditioned Embedding Diffusion for Text Generation. Strudel et al. 2022. \
[2] Analog Bits: Generating Discrete Data using Diffusion Models with Self-Conditioning. Chen et al. 2022. \
[3] TESS: Text-to-Text Self-Conditioned Simplex Diffusion. Mahabadi et al, 2024.

**Limitations:**

* The authors note that fully pretrained diffusion models are sparse, and that most diffusion language models remain at the small parameter regime (GPT-2).
* Limitations are adequately addressed.

---

> ### Author Rebuttal · Authors · 2024-08-06
>
> We sincerely thank Reviewer abfr for the review and are grateful for the time you spent with our submission. We wish to address your confusion and concerns by providing detailed responses to each of your comments.
>
> **Weakness 1: Simple datasets**
>
> The reasoning ability contains arithmetic reasoning and logical reasoning. From the experimental results, we validated that DoT performs as well as GPT models after fine-tuning on the boolean logic dataset. Yet, arithmetic reasoning such as grade school math is a more challenging task for all models. Besides, by introducing the relatively simple datasets, we aim to demonstrate that **DoT not only performs well on relatively simple tasks but also exhibits a higher efficiency compared to GPT models**.
>
> **Weakness 2: Throughput**
>
> Thank you for bringing up this point. When we show the throughput of DoT is superior to AR-CoT, what we want to emphasize is not purely throughput but its flexibility. As a hyperparameter, diffusion timesteps are determined through a held-out validation set. The number of time steps is influenced by the complexity of the task at hand. For instance, in simpler tasks such as digit multiplication, only a small number of timesteps is sufficient to obtain desirable performance. However, in more challenging reasoning tasks like GSM, we can increase the number of timesteps to enhance the performance. In this case, the final throughput is not superior to AR-CoT but we can achieve better results. In other words, we can spare more time on “thinking” on complex tasks (this interesting idea is introduced in Section 4.4 and Figure 4a). **We argue that this flexibility is exactly the advantage of DoT over auto-regressive CoT models**.
>
> Moreover, AR models can utilize key-value caching to enhance throughput during generation, but they still decode token-by-token for longer outputs, whereas diffusion models operate with fixed timesteps for longer context. Also, by removing kv-caching memory, the inference of diffusion models can be conducted with larger batch size, as mentioned in the SEDD paper [1]. In our paper, we highlight the potential efficiency advantage of DoT in our tested cases, and we believe that the efficiency of diffusion models in the wild is another interesting topic and worthy of further investigation.
>
> [1] Discrete Diffusion Modeling by Estimating the Ratios of the Data Distribution, ICML 2024.
>
> **Weakness 3: novelty of the methodology**
>
> In terms of various approaches that can be used in the fine-tuning stage of diffusion models, DoT is non-trivial. Other alternatives include directly fine-tuning Plaid 1B and using back-gradients to control the generation, and directly fine-tuning DoT models initialized with GPT2 parameters as shown in the Appendix. Compared with them, **the current DoT model stands out as the most effective. These empirical findings have not yet been addressed in any other existing literature**.
>
> Moreover, we propose the multi-pass variant of DoT and two sampling strategies to further enhance the performance, which we believe are significant in the diffusion realm.
>
> Finally, our paper also extensively discussed **the potential advantages of DoT over autoregressive models particularly for reasoning tasks**, such as the reasonability-efficiency trade-off, and the self-correction, which have not been explored to the best of our knowledge.
>
> **Q1: Connection between the first sentence of Section 3.2 and Table 2**
>
> Thanks for pointing out this potential confusion. The Plaid model is based on the continuous embedding space, and the model relies on gradient-based guidance to control the token generation. If we briefly follow this model to do the continue-training on the fine-tuning datasets (first row in Table 2), the performance is poor and we believe it is because the gradient-based guidance fails to do accurate conditioning. This motivates us to use DiffuSeq-style training. Please refer to the comments for detailed examples.
>
> **Q2: Sampling temperature**
>
> In our experiment, we found that enabling softmax temperature would slightly decrease the quality compared to greedy decoding (accuracy gap within 1%). But the diversity can be enhanced. As a result, we find a noticeable performance boost (4% on Plaid and 12% on SEDD) after performing self-consistency marginalization. We set the softmax temperature to be 0.5 according to the validation set in experiments.
>
> **Q3: Variable lengths**
>
> Yes, we use [PAD] to control the output length. The model will generate 128 tokens in total but all the latter [PAD] will be removed.
>
> **Q4: Connection between schedule sampling and self-conditioning**
>
> Thank you for bringing up this great question. The similarity between our schedule sampling and self-conditioning is that they both consider the condition of the model predicted sequence. However, they serve different purposes and are complementary in nature. The goal of self-conditioning is to use the previously estimated $\tilde x_0$ as additional feature besides the original $x_t$, so the network models $f(x_t,\tilde x_0,t)$, where $x_t$ is always corrupted from the oracle data.
> While the goal of our scheduled sampling is to add inference-time noise to $x_t$ to be consistent with the inference stage. So the network learns to model $f(\tilde x_t,t)$. **They are based on different purposes, and can be used together.** For example, we can potentially model $f(\tilde x_t,\tilde x_0, t)$ by providing $\tilde x_0$ as an additional feature.
>
> **Q5: Details for throughput results**
>
> For GPT models, we use KV-caching when decoding. For all models, considering both the small model size and context size, we didn’t use flash-attention.
>
> **Q6: Details for GSM-Aug dataset**
>
> Following the dataset setting in paper implicit-CoT, we keep the natural language in problem description but remove the natural language in the CoT response and only keep symbolic expressions in `<<>>`.

---

> > ### Author Response · Authors · 2024-08-08
> > **Detailed examples for Q1**
> >
> > Below we show an example on grade school math as a demonstration, where **bold** words in the query part are incorrectly recovered. We can see there are four recovered query tokens that exhibit minor differences due to soft gradient guidance, causing interference with the model's comprehension of the problem. That’s why we resort to hard control with gradient-free conditioning. We will add more details to clarify this confusion in the final version.
> >
> > >Groundtruth: Two trains leave San Rafael at the same time. They begin traveling westward, both traveling for 80 miles. The next day, they travel northwards, covering 150 miles. What's the distance covered by each train in the two days? <<2*80=160>> <<150*2=300>> <<300+160=460>> <<460/2=230>> #### 230
> >
> > >Prediction: **Three** trains leave San **Juan** at the same time. They **start** traveling westward, both traveling for 80 miles. The next day, they travel **southward**, covering 150 miles. What's the distance covered by each train in the two days? <<3*80=180>> <<180+80+150=340>> <<340/ 30=12.5>> #### 12.5

---

> > > ### Author Response · Authors · 2024-08-10
> > >
> > > Dear Reviewer abfr,
> > >
> > > Thank you for your valuable time to review our work and for your constructive feedback. We posted our response to your comments two days ago, and we wonder if you could kindly share some of your thoughts so we can keep the discussion rolling to address your concerns if there are any. If you have any further questions, we are happy to discuss them!
> > >
> > > Best regards,
> > >
> > > Authors

---

> > > > ### Author Response · Authors · 2024-08-14
> > > >
> > > > Dear Reviewer abfr,
> > > >
> > > > Thank you for your valuable time to review our work and constructive feedback. As the discussion period draws to a close, we would appreciate it if you could kindly take a look at our response to your comments. If you have any further questions, we are happy to discuss them!
> > > >
> > > > Thanks very much!
> > > >
> > > > Best regards,
> > > >
> > > > Authors

---

> > > > > ### Author Response · Authors · 2024-08-14
> > > > >
> > > > > Dear Reviewer abfr,
> > > > >
> > > > > As the discussion period draws to a close, we would like to emphasize the main point of our paper for your reference. Most reasoning paradigms currently rely heavily on CoT with autoregressive language models. The question remains as to whether diffusion language models are capable of reasoning like AR with CoT or surpass the performance of AR with CoT in efficiency or performance. In this work, we introduce DoT as an initial exploration in this direction and demonstrate the promising potential of diffusion models for reasoning tasks.
> > > > >
> > > > > Best regards,
> > > > >
> > > > > Authors

---

### Decision · Program_Chairs · 2024-09-25

**Decision:**

Accept (poster)

**Comment:**

The paper introduces Diffusion of Thought, to incorporate Chain of Thought style reasoning of autoregressive models into diffusion based models.

The reviewers identified several strenghts including application of the method to both discrete and continuous diffusion models, application on both pretraining and finetuning, strong performance on multiple benchmarks, and speed ups on simple reasoning tasks.

The weaknesses and areas of improvements identified by reviewers are: clarity issues in the write-up, missing discussions with highly related work (referenced by several reviewers), missing baseline experiments, requiring task-specific fine-tuning for certain tasks, and the idea being incremental being similar to DiffuSeq-style masking applied to CoT training data.

After reading the paper, discussions and the author response, to me it seems like most of the concerns have been either addressed or non-major. Regarding novelty, I believe the paper's idea of using COT in the diffusion process is very interesting and novel, and while there are still performance gaps with frontier AR models, the paper can lay foundation for future extensions in this work. I agree with reviewers that the experiments are comprehensive and explore various settings of applying DOT style training.
The next revision should ideally incorporate the reviewer feedback and rebuttal, expand on discussions of related work, and improve clarity of the paper, which should be doable without requiring major changes.